

# Aircraft and ground measurements of dust aerosols over the West Africa coast in summer 2015 during ICE-D and AER-D

Dantong Liu[1], Jonathan W. Taylor[1], Jonathan Crosier[1], Nicholas Marsden[1], Keith N. Bower[1], Gary

Lloyd[1], Claire L. Ryder[2], Jennifer Brooke[3], Richard Cotton[3], Franco Marenco[3], Alan Blyth[4],

Zhiqiang Cui[4], Victor Estelles[5], Martin Gallagher[1], Hugh Coe[1] and Tom W. Choularton[1]

[1] Centre for Atmospheric Sciences, School of Earth and Environmental Sciences, University of
Manchester, Manchester, UK

[2] Department of Meteorology, University of Reading, Reading, UK

[3]Met Office, Exeter, UK

[4]School of Earth and Environment, University of Leeds, Leeds, UK

[5]Dept. Fisica de la Terra i Termodinamica, Universitat de Valencia, C/ Dr. Moliner 50, 46100
Burjassot, Spain



## Abstract

During summertime, dust from the Sahara can be efficiently transported westwards within the Saharan Air Layer (SAL). This can lead to high aerosol loadings being observed above a relatively clean marine boundary layer (MBL) in the tropical Atlantic Ocean. These dust layers can impart significant radiative

effects through strong visible and IR light absorption and scattering, and can also have indirect impacts by altering cloud properties. The processing of the dust aerosol can result in changes in both direct and indirect radiative effect, leading to significant uncertainty in climate prediction in this region. During August 2015, measurements of aerosol and cloud properties were conducted off the coast of West Africa as part of the ICE-D and AER-D campaign. Observations were obtained over a 4-week period using the FAAM Bae146

aircraft based on Santiago Island, Cabo Verde. Ground based observations were collected from Praia, also located on Santiago Island. The dust in the SAL was mostly sampled in situ at altitudes of 2-4 km, and the potential dust age was estimated by backward trajectory analysis. The particle mass concentration (at diameter d=0.1-20µm) decreased with transport time. Mean effective diameter ($D_{eff}$) for super-micron SAL dust (d=1-20µm) was found to be 5-6µm regardless of dust age, whereas submicron $D_{eff}$ (d=0.1-1µm)

showed a decreasing trend with longer transport.

For the first time, an airborne laser-induced incandescence instrument (the single particle soot photometer, SP2) was deployed to measure the hematite content of dust. For the Sahel-influenced dust in the SAL, the observed hematite mass fraction of dust ($F_{Hm}$) was found to be anti-correlated with the single scattering albedo (SSA, λ=550nm, for particles d<2.5µm); as potential dust age increased from 2 to 7 days, $F_{Hm}$

increased from 2.5% to 4.5%, SSA decreased from 0.97 to 0.93 and the derived imaginary part ($k$) of the refractive index at 550nm increased from 0.0015 to 0.0035. However the optical properties of Sahara-influenced plumes (not influenced by the Sahel) were independent of dust age and hematite content with SSA ~ 0.95 and $k$ ~ 0.0028. This indicates that the absorbing component of dust may be source-dependent, or that gravitational settling of larger particles may lead to a higher fraction of more absorbing clay-iron

aggregates at smaller sizes. Mie calculation using the measured size distribution and size-resolved refractive indices of the absorbing components (black carbon and hematite) reproduces the measured SSA to within ±0.02 for SAL dust, by assuming a goethite/hematite mass ratio of 2. Hematite and goethite constituted 40-80% of the absorption for particles d<2.5µm, and black carbon (BC) contributed 10-37%. This highlights the importance of size-dependent composition in determining the optical properties of dust, and also the

contribution from BC within dust plumes.





# 1 Introduction

Saharan dust in Northern Africa contributes the majority of the aerosol optical depth (AOD) and produces
significant uncertainty in the prediction of climatic effects in this region. The Saharan atmospheric boundary
layer (SABL) can extend up to 6km from the surface in the summer which is the deepest boundary layer in
the world (Gamo, 1996). In summer, the Sahara Heat low (SHL) at low level caused by strong solar heating
moves northward, associated with high pressure at high/medium altitude located over the Saharan region.
The high pressure system enhances the persistent African easterly jet and efficiently advects the dry soil dust
over the African continent westward across the tropical Atlantic, resulting in a complex structure of stratified
dust layers at 3-6km in altitude - the Saharan air layer (SAL). The SAL can be transported thousands of
kilometres westwards towards the Caribbean basin (Tsamalis et al., 2013;Knippertz and Todd, 2012). This
hot, dry, sometimes dust-rich SAL creates a stable layer and overlies the cooler, more-humid surface of the
tropical Atlantic marine boundary layer (MBL), and the resulting strong temperature inversion may often
suppress the convection originating in the marine layer (Wong et al., 2009). At low level, the transport of
Saharan dust to the eastern tropical Atlantic is significantly controlled by the Azores high pressure system,
which leads to persistent northeasterly trade winds along the coast of north-western Africa (Carpenter et al.,
2010). Periodically the collapse of boundary layer can result in the transport of dust from the African
continent to low levels, while the north-easterlies along the coast of northwestern Africa can also provide a
pathway to entrain dust into the surface layer (Schepanski et al., 2009).

The dust-rich SAL alters the atmospheric radiation balance by interacting with both the solar shortwave and
terrestrial longwave radiation. The radiative absorption of dust heats the atmosphere and this heating further
intensifies the stability of the SAL (Dunion and Velden, 2004). By reflecting downwelling solar radiation,
dust can decrease the surface temperature over the ocean and land. The extent to which atmospheric
dynamics are altered by the SAL forms one of the largest uncertainties for the evaluation of climatic impact
in this region (Lavaysse et al., 2011). The dust could potentially impact the ice cloud microphysics within
this region as it may provide efficient IN (Richardson et al., 2007), and may affect tropical cyclone
development by modifying sea surface temperatures (Evan et al., 2008). During the westward transport, the
iron-rich dust could deposit into the ocean, supplying nutrients and affecting biogeochemical cycles in the
eastern tropical Atlantic (Jickells et al., 2005).

The size distribution, composition and spatiotemporal distribution of dust particles included in weather and
climate models is crucial for the correct application of these models. Firstly, the particle size determines the
distance over which dust is transported and its deposition rate, and hence its impact on regional climate
through changes in atmospheric circulation, surface air temperature and precipitation. The size of dust is
likely decreased during transport due to gravitational settling, when the larger and heavier particles tend to be
preferentially deposited. The challenges for models seeking to capture this process include determining the
source profiles of dust, simulating the emitted size distribution and mass loading, the size-resolved removal
rate and the vertical transport associated with convection. Previous studies investigated the dust size





distribution at different locations away from the source. Though there tends to be a consensus that the particle size is larger in close proximity of source (Weinzierl et al., 2009;Formenti et al., 2011;Ryder et al., 2013b) the size shift during transport is still an open question and is associated with the  properties of the dust and of the atmospheric dynamics (Formenti et al., 2011). Secondly, since the composition of the dust is

highly source dependent, the identification of the source of the dust is crucial to determine its properties at the receptor location. In particular, the free iron content such as hematite/goethite contained in Saharan dust crucially determines its absorbing properties (Zhang et al., 2015). Previous studies have used offline analysis of filter measurements to determine the iron content using X-ray diffraction, and have found a diversity of iron content across different dust source regions (Linke et al., 2006), and the offline composition results have

been associated with variations in the optical properties of dust (McConnell et al., 2010). The single particle information of dust has been long established through electron microscopy (Kandler et al., 2007;Krueger et al., 2004). A recent study demonstrated that the hematite/goethite in Saharan dust was embedded in the matrix of clay as fine grains with a variety of sizes (Jeong et al., 2016). There have been intensive aircraft measurements to characterize the dust size distribution and optical properties along the western African coast

(McConnell et al., 2010;Haywood et al., 2003;Ansmann et al., 2011;Weinzierl et al., 2012) and over the African continent (Haywood et al., 2008;Ryder et al., 2013b;Lebel et al., 2010;Osborne et al., 2011) in the last decade, however the absorbing iron component in the dust, which importantly determines its radiative forcing, has not been characterized by an in-situ measurement.

The Cabo Verde region off the coast of western Africa serves as an ideal location for the in-situ

measurement of dust outflow properties. The dust in the SAL over Cabo Verde is transported from a diversity of soil sources across the north African continent and is ideal for the study of atmospheric processing of dust. In August 2015, an intensive multi-platform field campaign involving both aircraft and ground measurements was conducted. The Ice in Cloud Experiment - Dust (ICE-D) project focussed on the development of tropical convection in which aerosol-cloud interactions are suspected as having a strong

influence on the evolution of convective clouds (Mahowald and Kiehl, 2003;Twohy et al., 2009). It was carried out in conjunction with the AER-D (AERosol properties - Dust) and SAVEX-D (Sunphotometer Airborne Validation EXperiment in Dust) projects investigating the properties of dust in the SAL above the Eastern Atlantic. Fig. 1 shows all of the flight tracks associated with this joint study. All the flights took place within the close vicinity of the Cabo Verde Islands, with the exception of flights B923 and B924 which

sampled thick dust much further north (around 24°N). The properties of aerosols in this region deposited on the ground were also investigated through the month long ground-based measurement experiment at Praia Airport, Santiago Island. These experiments provide a comprehensive dataset to investigate the SAL dust properties during the summer season.



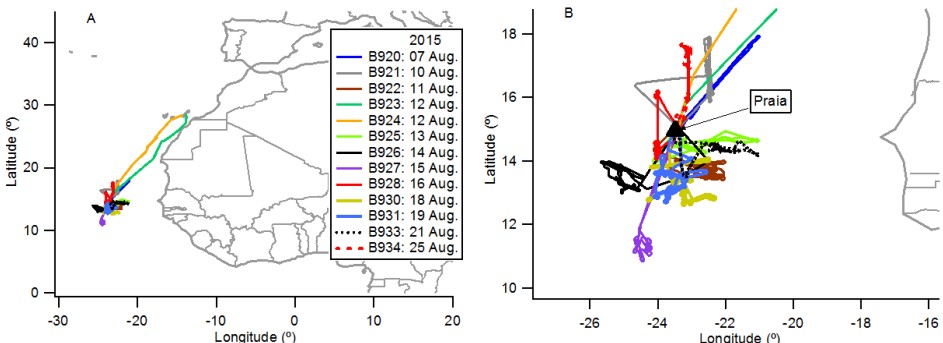

Fig. 1. A) A summary of all flight tracks. B) zoomed in view of flights tracks, and the solid triangle denotes the location of the ground measurement site.

## 2 Flight summary, instrumentation and data analysis

### 2.1 Overview of the instruments

The ICE-D and AER-D flights took place during Aug. 2015. The measurements described here were made using the Airborne Research Aircraft (ARA), a BAe-146 aircraft operated by the UK Facility for Atmospheric Airborne Measurements (FAAM), over Cabo Verde off the western Africa coast. A number of straight and level runs (SLRs) and profiles were performed during each flight. The instruments employed on the aircraft enabled a range of measurements of aerosol size distribution, chemical composition and optical properties. The wing-mounted Passive Cavity Aerosol Spectrometer Probe (PCASP, Osborne et al., 2008) measured optical size from 0.1-2.5µm. It is assumed to sample at 100% efficiency at all sizes because of the short sampling lines and relatively small maximum size of particles sampled. The uncertainties of PCASP-measured size distribution due to non-spherical particle shape (Osborne et al., 2008) are moderate compared to other sources of uncertainty affecting the optical sizing. The cloud droplet probe (CDP-100) measures larger particles with diameters of 5-40µm. This paper focuses on size distributions up to 20um diameter for a convenient comparison with previous studies. Larger aerosol particles up to ~100 um were observed in this campaign, however, the analysis of these coarse and giant particles is provided in a separate paper by Ryder et al. (in preparation). Full calibrations of both the PCASP and CDP were carried out before and after the campaign (and checked prior to each flight). Size bins were determined based on a refractive index (RI) 1.53-0.001i which was previously shown to be appropriate for West African dust measurements (Rosenberg et al., 2012). The size distributions derived from the CDP and PCASP were combined to obtain the full size spectrum for aerosols from the wing-mounted instruments. Periods of sampled cloud were screened out before analysis using a threshold liquid water content measurement of 0.001 g m$^{-3}$ (Crosier et al., 2011).





Measurements of aerosol absorbing properties were made using a Radiance Research Particle Soot Absorption Photometer (PSAP). PSAP data are only used from straight and level runs (SLRs) where the instruments flow rate was manually set to 3lpm at standard temperature and pressure (STP, 273.15K and 1013.25 mbar). The PSAP data are corrected for shadowing effects on the filter and for multiple scattering

interactions according to the technical note released by the Met Office (Turnbull;Haywood and Osborne, 2000). PSAP filters were changed before every flight. Aerosol scattering properties were measured by a TSI integrating nephelometer model 3563 at three wavelengths (450, 550 and 700 nm) in dry conditions. The nephelometer measurements were corrected according to Anderson and Ogren (1998), assuming particles sized up to 2.5μm reach the nephelometer. The scattering Ångström exponent (SAE) is calculated between

λ=450 and 700nm, which is often used as a qualitative indicator of aerosol particle size or fine mode fraction (Seinfeld and Pandis, 2016).

The physical properties of individual refractory absorbing particles were characterized using a single particle soot photometer (SP2) manufactured by DMT Inc (Boulder, CO, USA). The instrument operation and data interpretation procedures of the specific Manchester SP2 instrument have been described elsewhere (Liu et

al., 2010;McMeeking et al., 2010). The SP2 incandescence signal was calibrated for BC mass using Aquadag® black carbon particle standards (Aqueous Deflocculated Acheson Graphite, manufactured by Acheson Inc., USA) and corrected to ambient BC using a factor of 0.75 (Laborde et al., 2012). The composition of the absorbing particles incandescing in the SP2 laser beam can be discriminated by the ratio of irradiance detected at different wavelength bands (as detailed in section 2.3). The SP2 determines a single

particle to be absorbing if the incandescence signal is above a threshold, otherwise it is considered to be a non-absorbing scattering-only particle.

The chemical composition of non-refractory PM1 was measured by an Aerodyne C-ToF aerosol mass spectrometer (AMS). A detailed description of the instrument can be found elsewhere (Drewnick et al., 2005;Morgan et al., 2010). A composition dependent collection efficiency (CE) was applied to the data

based on the algorithm by Middlebrook et al. (2012). The AMS was calibrated using mono-disperse ammonium nitrate particles. All the SP2 and AMS measured concentrations are reported as mass concentrations at standard temperature and pressure (STP, 273.15K and 1013.25 mbar).

The collection efficiency of the Rosemount inlet used on the aircraft to deliver aerosols to internal instrumentation is characterized as being >80% for particles smaller than 2.5μm (Trembath et al., 2012). The

aerosol optical and composition measurements in the cabin, therefore, are representative of the accumulation mode and smaller aerosols only (i.e. for d<2.5um).

Ground based ambient aerosol measurements were made in the mobile Manchester Aerosol Laboratory located within the perimeter of Praia International Airport, Santiago Island, Cabo Verde (14°57'N 23°29' W, 100 m asl), approximately 1500m from the coast and 150m from the airport runway. A pumped inlet system

with a size cut off of 10μm was fixed to a sampling height of ~10 m a.s.l., and pulled a total flow of ~1000 L



min$^{-1}$ down the inlet. A series of online instruments were distributed downstream of the inlet. The instruments included size distribution measurements from a Grimm Technik 1.129 Sky Optical Particle Counter (OPC, Heim et al., 2008), over the range 0.26-30µm an aerodynamic particle sizer (APS) model 3321(TSI, Inc., St. Paul, MN) over the range 0.5-20µm and a TSI Scanning Mobility Particle Sizer for

ultrafine particle size distribution measurements. For comparison, a second Grimm measured in-situ on the roof of the container to avoid any of the inlet losses associated with sampling down the 10m inlet line. A laser ablation aerosol particle time-of-flight (LAAP-TOF) single-particle mass spectrometer (Marsden et al., 2016) was deployed to measure the composition of refractory aerosol in the size range 0.5-2.5µm. Other instruments included a WIBS-4 bioaerosol spectrometer (Gabey et al., 2010) for biological aerosol detection,

and aerosol filter samples were also collected for offline compositional particle analysis (these and related measurements will be reported elsewhere).

## 2.2 The aerosol size distribution

Fig. 2 gives examples of aerosol size distributions from all instruments used in this study. The PCASP can optically size particles up to 2.5µm. The size distribution of the SP2 includes both incandescence and

scattering-only particles. The SP2 is not able to size particles d>0.65 µm due to the saturation of the scattered light detector (shown as the hump in the last bin of the blue line in Fig. 2), however the counts for large saturated particles are not affected. To validate the discrepancy between wing-mounted measurements and those in the cabin, the particle counts from SP2 and PCASP are compared across the overlapping size ranges at both 0.18-0.5 µm and 0.5-2.5 µm, termed the _small or _large modes respectively. As Fig. S1 in

supplement shows, the PCASP counts correlate with the SP2 counts across the smaller mode, but the PCASP tends to measure more particles in the large mode which may be due to the reduced aspiration efficiency of the SP2 inlet and effects of the aircraft sampling line for larger particles. At higher altitudes the PCASP/SP2 ratio is more frequently lower than unity which may also result from the concentration enrichment of the Rosemount inlet at high aircraft speed; whereas at low level, the diffusion losses in the sampling line could

cause the SP2 counts to be lower than the PCASP. This altitude-dependent PCASP/SP2 ratio is used throughout to enable comparison between wing-mounted and cabin aerosol measurements.

For each SLR, the effective diameter ($D_{eff}$) is calculated. It is defined as the ratio of the integrated volume and integrated area from the averaged size distribution (McFarquhar and Heymsfield, 1998), as expressed in Equation 1,

$$D_{eff} = \frac{\int D_p^3 \frac{d_N}{dDp} dDp}{\int D_p^2 \frac{d_N}{dDp} dDp} \qquad (1),$$

$D_{eff}$ is calculated for both the submicron mode (0.1-1µm) and the supermicron mode (1-20µm) respectively, from the one-minute averaged size distribution, and its variability is given over the straight run or sampling period.





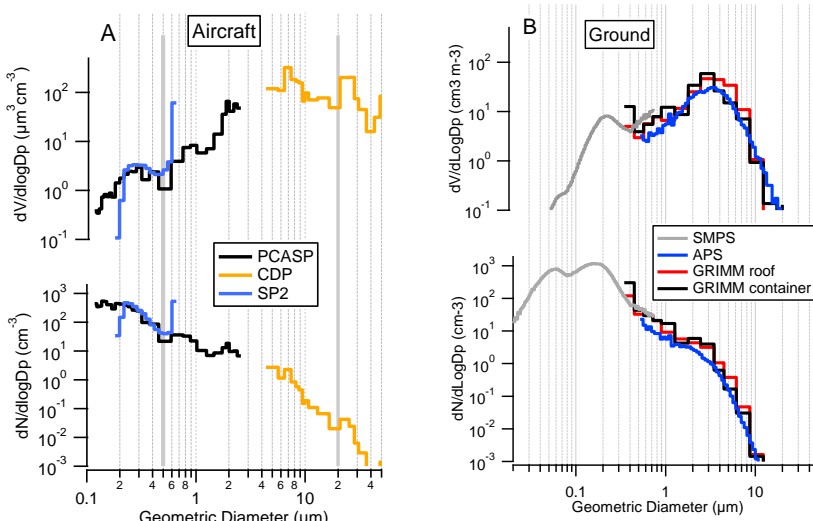

Fig. 2. A typical example of size distributions for B928 SLR4 (see Fig. 7) and ground experiment Period I (see Fig. 10).

For the ground experiments, the GRIMMs on the roof (without inlet aspiration effects) and in the container

measured simultaneously. The high consistency between the instruments indicates only minor aerosol loss through the sampling line of the container at d<10µm. To be consistent with aircraft measurement, the sub-micron $D_{eff}$ is calculated over the same range (0.1-1µm), but for the supermicron mode it is calculated over a larger range (1-10µm) which is within the detectable size range of the instruments and sampling line cut off size.

2.3 The hematite content of dust aerosol

The SP2 can detect any refractory particle which is sufficiently absorbing to incandesce at 1064nm. Both BC and hematite can absorb, but will exhibit different intensity of irradiance at different bands of visible wavelength when incandescing. The SP2 measures at two different wavelength bands (one is broad broadband at 470-850nm, the other is narrowband at 650-870nm) and so can discriminate different absorbing

components using the ratio between both bands, known as the colour temperature ratio ($T_c$). Fig. 3 shows the distributions of $T_c$ as a function of broadband incandescence signal. Note that the incandescence signal is proportional to the particle mass for each composition, but the relationship between the incandescence signal and particle mass needs to be specifically calibrated for each different absorbing component. The $T_c$ for lower incandescence signals ($<1e^5$) shows up as a single mode, however for larger incandescence signals $T_c$

values appear in a second lower $T_c$ mode. We have compared results from a wide range of BC-only environments such as diesel chamber experiments, urban traffic/solid fuel burning and open biomass burning studies (Supplement Fig. S2), and these show that the lower mode of $T_c$ values is never present when sampling BC. The hematite measurement by the SP2 in the laboratory also shows behaviour consistent with





the lower $T_c$ distribution. For Saharan dust, hematite and goethite are considered to be the principal compositions of absorbing free iron (Zhang et al., 2015). Laboratory experiments of goethite show no detectable signal from the SP2 due to its very weak absorption at 1064nm. Given this, we assume that hematite is the main absorbing component of the dust observed using this technique. On the ground, the SP2

measured hematite is highly correlated with the number concentrations of particles classified as silicate from clustering of single particle mass spectra from LAAP-TOF measurements (Fig. 10E, as detailed in (Marsden et al., 2016)), suggesting the measured hematite is a valid tracer for dust. Also, as Fig. 5 shows, for measurements within the MBL, the aerosol mass (d=0.5-20μm) derived from PCASP and CDP exhibits significant spikes, which is probably the influence of sea-salt, whereas the SP2-measured hematite is

unaffected by sea-salt and so tends to be a better tracer of dust.

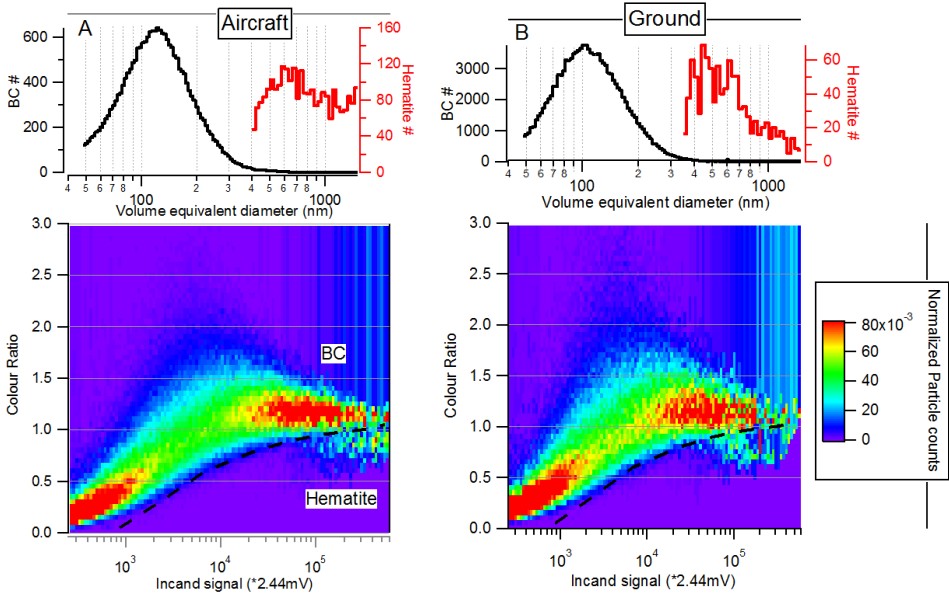

Fig. 3. The colour temperature distribution as a function of broadband incandescence signal as coloured by particle number density, for aircraft (A) and ground measurements (B) respectively. The particle density is normalized by the total particle number at each incandescence signal bin. The top panels show the extracted

BC and hematite numbers according to the dashed line on the image plots (lower panels).

The calibration of hematite was performed by suspending a sample of pure composition in a tube. The particle mass is selected directly by a Centrifugal Particle Mass Analyser (CPMA, (Olfert and Collings, 2005) and the SP2 measures the mass-selected particle (Liu et al., 2017). The hematite mass is then derived from the incandescence signal based on this calibration and is converted to a mass-equivalent diameter. Note

that the detection efficiency declines for smaller hematite particles (Supplement Fig. S3), and this factor is scaled up for the derived size distribution. Fig. 3 shows results from typical examples of dust plumes from aircraft and ground measurement. The hematite size is larger in the aircraft than on the ground, peaking at





about 0.65µm and 0.45µm respectively. The same mass of hematite may be included within different sizes of individual dust particles. The scattering signal is saturated for almost all of the hematite-containing particles resulting in a lack of capability for the SP2 to resolve the hematite fraction within single particles, in other words the particle size that contains hematite is unknown. In reality, submicron grains of iron (hydr)oxides

(goethite and hematite) are commonly dispersed in the clay-rich medium in Saharan dust (Lafon et al., 2006;Kandler et al., 2009). The often co-existing materials with hematite in real dust particles may prevent or limit its incandescence thereby decreasing detection efficiency, which may lead to an underestimation of hematite in the small dust size range. There is therefore less confidence in the SP2 derived total number concentration of hematite containing particles. However, smaller hematite particles are not optically

important and the SP2 is considered able to detect the majority of the hematite mass.

As the hematite-containing particles all saturate the scattering detector of the SP2, (which occurs when particle size >0.5µm), hematite is considered to be only contained in larger particles >0.5µm. Particles larger than 0.5µm are also deemed to be the main size mode of dust particles according to the off-line size-resolved composition analysis (Kandler et al., 2009;Kandler et al., 2011b). Though there may be a number of dust

particles smaller than 0.5µm, small hematite containing particles may not be significantly detected due to the reduced detection efficiency of such particles. Due to the inability of the SP2 to size the large particles, the PCASP measured size in the large mode is converted to a volume distribution to work out the hematite volume fraction in the large particle ensemble (at 0.5-2.5µm), using the previously derived correlation between PCASP and SP2 (Fig. S1). Note that because goethite is not able to incandesce under the SP2 laser,

the hematite fraction reported here is only part of the free iron fraction. We note that the goethite content was previously found to contribute about 50-75% of free iron for the Sahara dust ((Zhang et al., 2015), and references therein).

## 3 The meteorology and classification of dust source regions

Fig. 4 shows a 4-day sequence of 700mbar geopotential height and wind field plots for the region. This is the altitude at which the dust plumes were mostly observed. There was a change in the synoptic conditions on 15/08, after which the mid-altitude high pressure centre moved from the northwest to over central Sahara, leading to a stronger easterly wave over the Cabo Verde region. This is consistent with the in-situ aircraft wind (and thermodynamic) measurements as shown in supplement Fig. S4, with flight B927 onwards

showing a high horizontal wind speed >10m/s at the level of dust plumes. The eastern Atlantic marine boundary layer (MBL) is about 300-500m deep based on the potential temperature ($\theta$) profile. There is clear wind shear at the top of the MBL (Fig. S4). Within the MBL the wind direction is predominantly N or NE and the horizontal wind speed is consistently about 5m/s, hence, the movement of air within the MBL is slow and constant. The movement of air masses at low level is mainly controlled by the Azores anticyclonic

system located over the northern Atlantic, leading to persistent north-easterly winds, where the observed dust



involved in this slow low-level transport tends to have experienced considerable processing. Above the MBL, there is significant wind shear and the dry and warm SAL overlies the MBL, and here the wind speed is significantly higher and more easterly in direction. The wind speed and direction in the SAL depends on the synoptic conditions: in the first half of the campaign from 10/08 to 14/08, the maximum horizontal wind

speed in the SAL was lower than 10m/s and the wind direction over Cabo Verde varied between NE and SE. This is consistent with the measured lower dust loadings in the SAL from B921-B926 shown in Fig. 5, when the SAL and hence higher dust loadings were in a more northerly location (note that B924 was conducted in a different location so should be isolated from the discussion here). Less than 1 mg m$^{-3}$ of aerosol mass loading was seen in the size range d=0.5-20μm and < 3 μg m$^{-3}$ was measured as hematite. During the second

half of the experiment, after 15/08, the centre of high pressure moved over the central Sahara and the Cabo Verde region was more intensively influenced by the African Easterly Jet and the SAL. The wind speed was significantly enhanced above 2km (at >10m/s) and wind direction more consistently from the east. In such conditions it is highly likely that stronger easterly winds might bring enhanced dust loadings from the African continent which is consistent with the observations of aerosol mass concentration during this period

which increased to 1-2 mg m$^{-3}$ (for d=0.5-20μm) of which 3-6 μg m$^{-3}$ was measured as hematite mass.

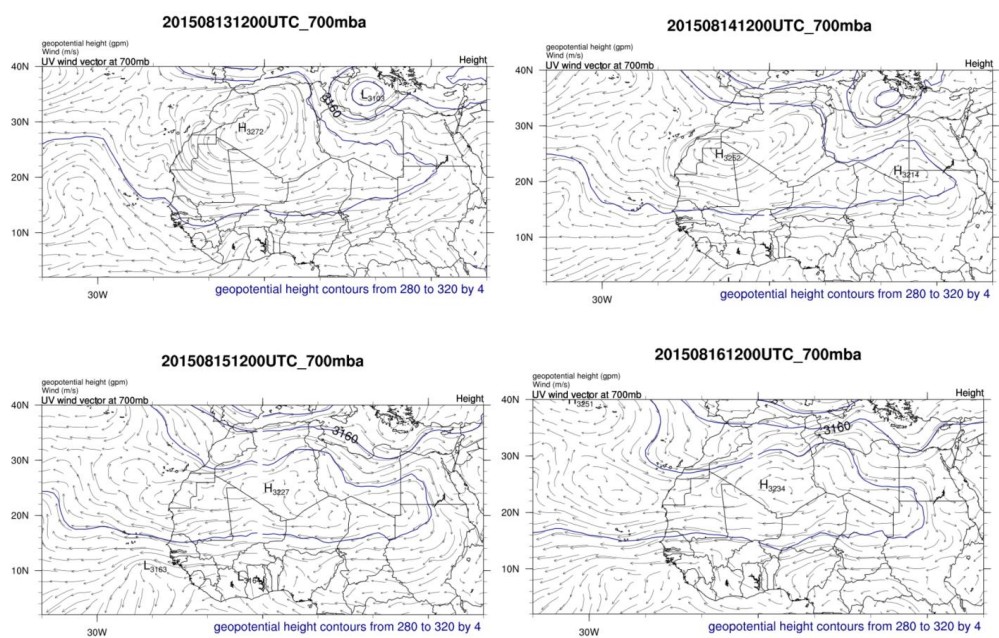

Fig. 4. The 700mbar geopotential height and wind field from 13/08 to 16/08 based on ECMWF reanalysis.





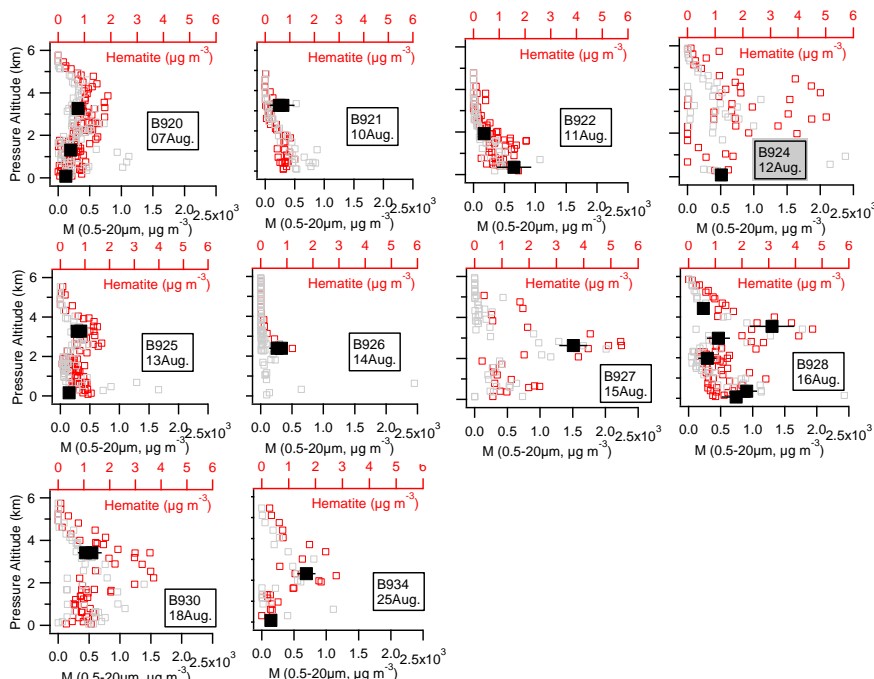

Fig. 5. The aerosol mass concentration (d=0.5-20µm) derived from PCASP and CDP size distributions and hematite loading from the SP2 in profiles. The black markers show the mean aerosol mass (d=0.5-20µm) derived during straight and level runs (SLRs) with error bars showing ± one standard deviation. The high level remote sensing flight (B923), the flight without the AMS running (B929), and the flights without aerosol profiles or SLRs (B931 and B933) or the fight without successful AMS measurements (B932) are excluded from the analysis here.

Back-trajectories (BT) of air parcel history and transport were created using the Hybrid Single-Particle Lagrangian Integrated Trajectory model (HYSPLIT) (Draxler and Hess, 1998). Real-time back-trajectories were calculated every 1min along the aircraft track and every 1h for ground measurements, and calculated 8 days backward in time. Horizontal and vertical wind fields for trajectory calculations were provided by the 1°×1°, 3-hourly GDAS1 reanalysis meteorology (Global Data Assimilation System; NOAA Air Resources Laboratory, Boulder, CO, USA). The different source regions and land types the air masses passed over are defined as the sea (SE), European continent (EU), the African continent which is separated into the Sahara Desert (SD) and Sub-Sahara Africa (SA) (using a border line as shown in Fig. 6), and finally the region local to Cape Verde (CV). The border line separating SD and SA is set at 17.5°N, broadly according to the soil inventory map below which line the Sahel region has dramatically different mineral soil compositions that





include a higher hematite content and lower feldspar content compared to the Sahara Desert (Nickovic et al., 2012). The border line between SD and EU is set at 34.5°N.

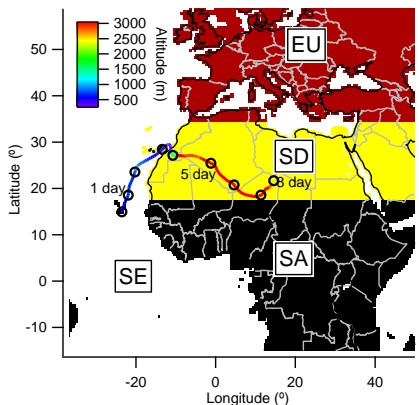

Fig. 6. An example back-trajectory (with end point at 12:00, 12[th] Aug. at the ground-based site) and calculation of transport days over various land surface types for this back-trajectory, separated into the defined land surface type regions.

Given the deep boundary layer over northern Africa in summertime, the assumption here is that the BTs passing over continental Africa encountered a homogenous vertical dust distribution, and that the air masses could transport dust from any range of sources along its pathway. According to the cross section sub-type aerosol product from CALIPSO (Winker et al., 2009) (Supplement Fig. S5), the dust was shown to be well mixed up to 5-6km over the majority of the Saharan continent. Below altitudes of 5km (a.s.l.) the BTs are thus deemed to be largely influenced by surface sources, and are assigned to a land type as shown in Fig. 6 for each hourly point along the BT as long as the altitude condition is met. We are therefore able to derive a fraction of BT points passing over each land type over the course of 8 days, as Fig. 7C shows. The air mass could spend a range of time over a specified land type and this range is represented by a standard deviation for each single BT. For the example shown in Fig. 6, the air mass has spent between 2-6 days over SE region which is represented as 4±2 days, and spent between 4-8 days over SD, represented as 6±2 days. This calculation is performed for every single BT for both aircraft and ground-based measurements. Combining all BTs gives the time series of air mass types and transport days. The overall residence time, here defined as transport time over each of the land types, can therefore be calculated (Fig. 7D).

The classification of each BT is performed according to the following criteria: the BT will be firstly assigned as SA type if the SA air mass fraction is >5%, and then if the SD mass fraction is >5% it will be assigned as SD. Air masses with transit time <5% from SA or SD are assigned as SE. Where the BT necessarily passed over SD for most of its time, but the SA air mass is designated, it is actually a mixture of the SA and SD air masses (thus termed as SA+SD). However, the SD air mass is thus defined only if it is solely influenced by SD (and not by SA). The air mass type for a whole SLR from aircraft measurements is then determined by





the majority of the BT air mass types within the SLR. For aircraft profiles, each point of the profiles (in 1min resolution) is assigned with an air mass type.

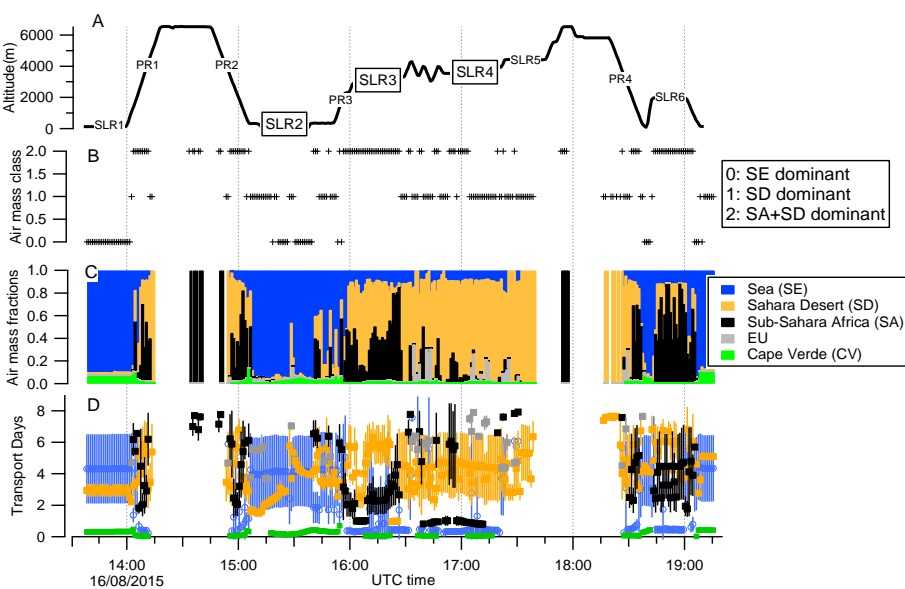

Fig. 7. The BT analysis for flight B928 on 16 Aug 2015, from top to bottom: A) the position all straight and level runs (SLR) and profiles (PR) marked on the altitude track; B) the assigned air mass type in one minute resolution; C) the air mass fractions for every minute; D) the transport days over each air mass source type.

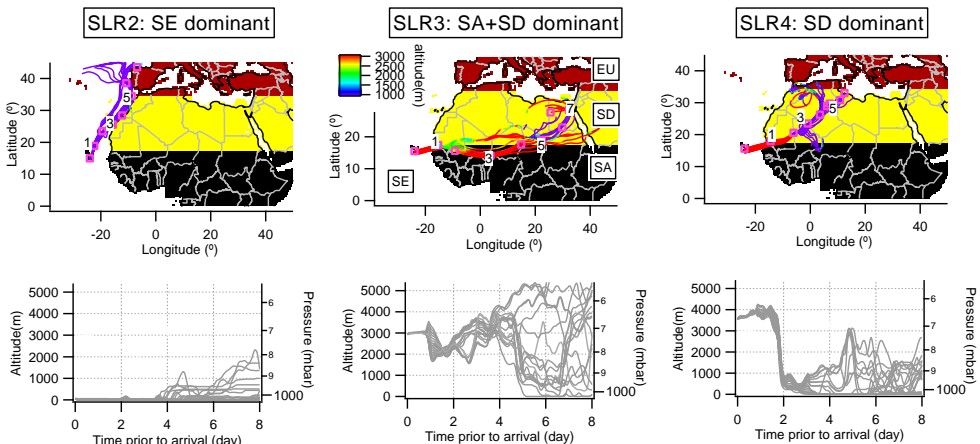

Fig. 8. The BTs for the SLRs marked in the boxes in Fig. 7A.



Fig. 7 gives an example of the BT analysis for flight B928 when the aircraft conducted a sequence of SLRs with influences from different air mass types. Three typical cases with SE, SA+SD and SD influences are shown, corresponding to SLR2, SLR3 and SLR4. CALIPSO aerosol type cross sections corresponding to 3, 5 and 7 days preceding the flight date are shown in Fig. S6. Note that the SE influence dominates at low level when flying in the MBL, with consistent north-easterlies along the coast of NW Africa. Additionally, many SE cases were designated at higher altitudes when dust plumes were observed (see Fig. 11). This is consistent with the Met Office forecast model which predicted high AOD (Fig. S5) in a region coincident with the back trajectory five days previously being close to the western coast of Morocco. A further CALIPSO cross section analysis for off the coast of Morocco (Fig. S6) shows the presence of a dust layer up to 4km which encompasses the BT altitudes. The transport time of dust for SE air masses cannot be estimated as the BT has not shown significant contact with any continental surface sources, as illustrated in the example shown by SLR2 in Fig. 7 and the left hand panel of Fig 8. Flights B925 and B926 are the main flights composed of SAL air but designated as SE air mass (Fig. 9). These flights were more southerly and closer to the ITCZ, coinciding with an elevated baroclinic dust layer with high moisture content (Carlson, 2016), where the main dust layers may have been spread out by significant wind shear and the BT analysis is thus unable to track their origin (Ryder et al., 2017, in prep).

For SLR3 in Fig. 7, designated as SA+SD, the air mass is largely easterly, the receptor is influenced by closer SA sources (spending under 4 days over SA) and more distant SD influences (with BTs spending up to 2 days over the SD region). Consistent with the Met Office forecast AOD product, most of the BTs have passed through the dust-laden region, though with some divergence after 5 days transport. The CALIPSO cross section (Fig. S6) indicates most of the BTs have passed through the deep boundary layer over the African continent, with a mixed dust layer up to 5-6km. The dust observed during SLR3 is therefore a mixture of dust sources along 17.5°N extending from western Mauritania to northern Sudan with any dust from eastern sources experiencing a longer transport time. SLR3 is therefore classified as SA+SD. Given the SD influence is less significant for this SLR due to the shorter time spent over SD, the potential dust age is therefore calculated to be the transport time over SA land of ~2 days subsequent to the measurement time, and time spent over the SE and SD region is excluded from the transport time. For SLR4 in Fig. 7, designated as SD, the air mass movement is north-easterly with the majority of the dust advected from the Western Sahara spending 2-6 days over the SD region. The dust transport time is calculated as the time over SD land, since this BT spent most of its time over the SD region (4 days), and other regions over which the BT spent time, such as SE, are not included in the transport time. Identical BT analyses for each SLR for the other flights are shown in supplement Fig. S7. Transport times shown in this study are essentially time spent by the BT (subsequent to measurement time) over the airmass region for which the BT spent the most time.

There are several advantages and limitations of this method of assigning regions of air mass influence and dust age. The advantages include the relatively high time and height resolution possible with which to initiate BTs using HYSPLIT, and to track airmasses therein. However, during Saharan summer the dominant uplift mechanism has been shown to be haboobs, uplifting 50% of dust, driven by convective outflows from



mesoscale convective storms (MCS) (Marsham et al., 2013). Since the meteorological reanalyses driving the HYSPLIT BTs are not able to resolve these convective events over West Africa, the BTs may not be able to identify either the dust source location, or potentially the transport pathways over these or subsequent dust uplift events (Sodemann et al., 2015). It is possible to utilize SEVIRI RGB satellite imagery to determine

dust source locations and uplift times (e.g. (Schepanski et al., 2007)) but this does not permit different altitudes to be resolved. Dust events observed during four of the flights here were examined using SEVIRI imagery by Ryder et al. (2017, in prep) and for all four cases MCS events and haboobs drove the dust uplift and subsequent transport. Therefore there is a level of uncertainty in the BT pathways and transport times since they do not capture these MCS events. Additional uncertainties stem from the assumption that dust

beneath 4km is influenced by the underlying surface for SA and SD regions. Dust may be inhomogeneously distributed in the vertical, and is likely to be subject to dry or moist convection and the daily redevelopment of a deep boundary layer (Trzeciak et al., 2017). The air mass may have received the dust deposited from upper layers from previous days as a result of sedimentation. Therefore the dust transported could be mixed with that from a range of surface sources along the BT. The analysis presented in this study is not attempting

to track the exact source origin of single dust plumes but to broadly evaluate the regions contributing most to dust events in the sampling region.

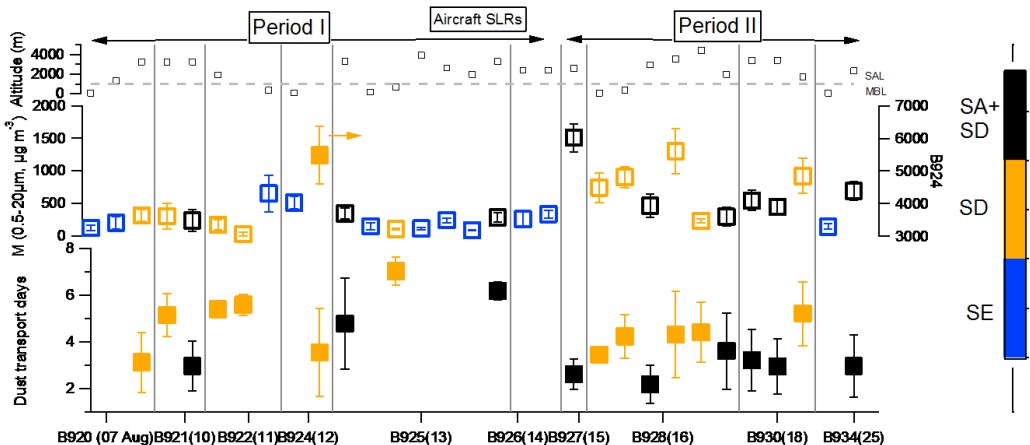

Fig. 9. A summary of altitude, potential dust age and aerosol mass (0.5-20µm) for all aircraft SLRs. Note the aerosol mass for flight B924 is shown on the right axis. The numbers in brackets show the date of flights in

Aug. The grey dash line indicates the layer height separating the SAL and MBL.

A summary of dust transport days (dust age) for all SLRs in this study is shown in Fig. 9. The dust age for SE cannot be identified by the BT analysis and thus is not shown here. Consistent with the synoptic conditions, dust plumes with SD origin were more frequently encountered during Period I (before 15$^{th}$ Aug.) whereas dust from SA+SD influenced air masses were more frequently observed during Period II when the

wind was more easterly. During period I, due to the lower wind speed, the dust experienced a longer




transport time (except for B924 when a strong dust storm was encountered along the coast of north-western Africa, exhibiting much higher dust loadings and a mixture of dust ages). From B926 onwards, the efficient transport of dust via a stronger easterly wind led to larger advected dust loadings and to reduced dust transport times.

An identical BT analysis was performed for the ground site at hourly intervals for a starting point 500m a.s.l. above the ground site (Marsden et al., 2016). As Fig. 10 shows, the ground site was mostly influenced by marine air masses, with the dominant features of summertime low-level circulation over northwestern Africa, characterized by a predominant north or northeasterly BT along the coast of Morocco, Western Sahara and Mauritania. The BT analysis on the ground clearly identifies two dust events with longer and shorter

transport time, shown as Periods I and II in Fig. 10 respectively. The dust from Period I originated from both the northwest coast of Africa and longer transport from the African continent with potential dust age about 5-7 days. The dust observed in Period II is dominated by constant transport from the dust-laden coast along Morocco and Western Sahara with a time scale of 2 days. Both pathways advected dust but from different source origins, and this is reflected in the different size and compositions of mineral dust in the two air

masses (Marsden et al., 2016).

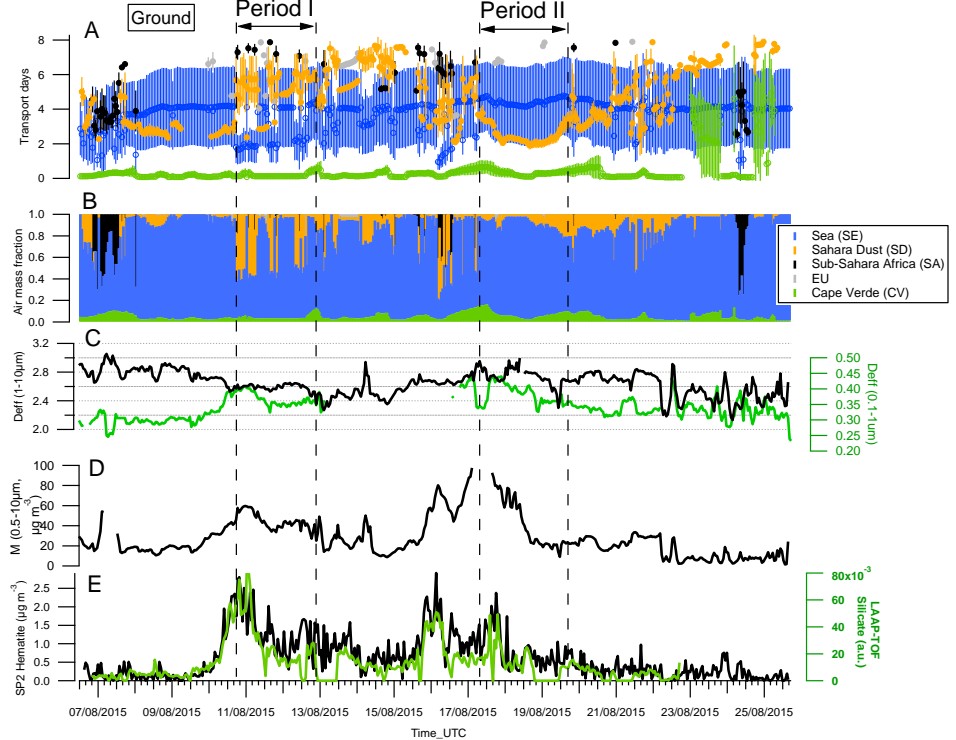



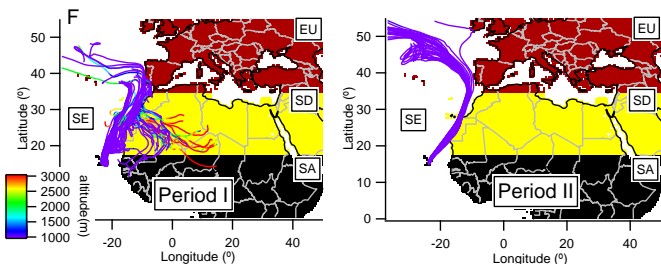

Fig. 10. The results from ground measurements: Time series of A) BT airmass-type transport days; B) BT airmass-type fraction; C) the effective diameter at submicron and supermicron size; D) aerosol mass (0.5-10µm); E) the SP2 detected hematite mass and LAAP-TOF detected number concentration of silicate-containing particles; F) the back-trajectory ensembles for Period I and Period II.

# 4 The physiochemical properties of dust and other aerosols

## 4.1 Compostion of aerosol

Fig. 11 shows the measured compositions for all aircraft straight and level runs (SLRs) and profiles as a function of the dominant BT classification region (note that the compositions reported here are for particles <2.5µm). In summertime, the vast majority of intensive open biomass burning occurs across Southern Africa (Hao and Liu, 1994) and so did not significantly influence the ICE-D measurements. This is consistent with the generally low loadings of organic matter ($<1µg\ m^{-3}$) and BC ($<0.5\ µg\ m^{-3}$) observed for all air mass origins; although some profiles/SLRs sporadically captured relatively high loadings of BC and OM (which were also well correlated), but it is unclear whether these plumes originated from domestic cooking sources or open biomass burning. The sulphate consistently showed a higher loading near the surface within the MBL and decreased loadings with altitude for profiles of all air mass types. The AMS derived ammonium and sulphate loadings ($NH_4$ and $SO_4$) had an equivalent ratio of 1.5-2 for most of the SLRs; highly acidic sulphate was also previously observed along the western coast of South America (Lee et al., 2014). There were some SLRs with relatively high sulphate and hematite loadings during SA+SD air masses. The formation of this sulphate may be related to the increased anthropogenic activities towards sub-Saharan Africa which has then been mixed with dust during transport.



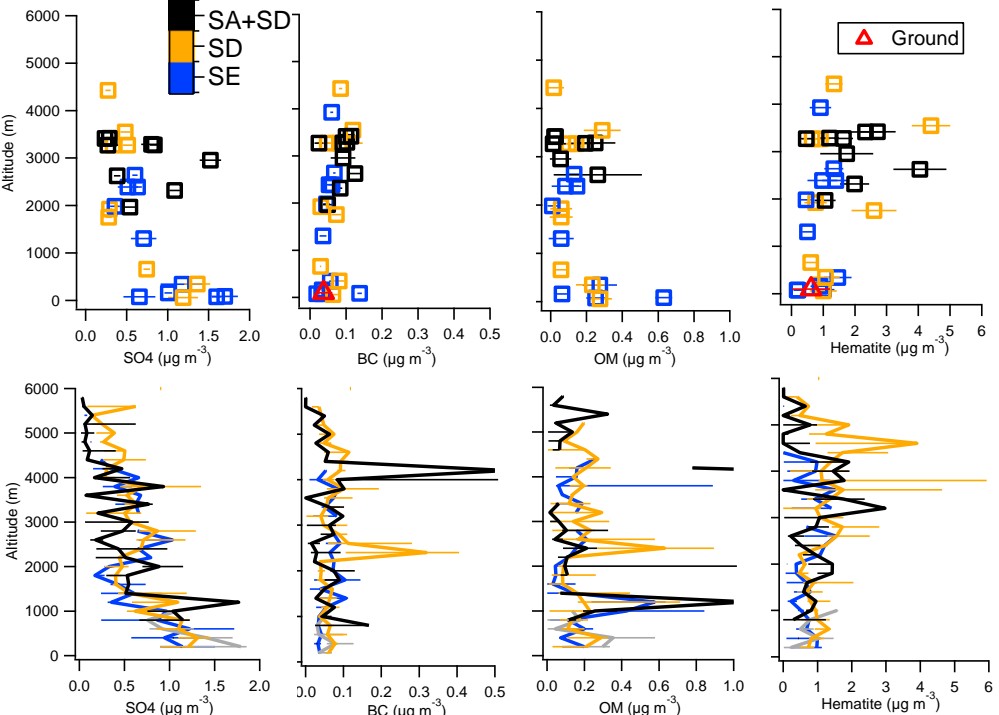

Fig. 11. Top panels (left to right) show the mean value and standard deviation of Sulphate (SO4, µg m$^{-3}$), Black carbon (BC, µg m$^{-3}$) Organic Mass (OM, µg m$^{-3}$) and Hematite (µg m$^{-3}$) from aircraft SLRs from different source origins (SE, SA and SA+SD). Bottom panels show the median value of vertical profiles for all flights (all of the profiles are grouped according to air mass type classification with 1 minute resolution), with the error bars showing the 75$^{th}$ and 25$^{th}$ percentiles. Note that the data from B924 are excluded because of this flights different location and anomalously high dust loadings which are off-scale in this figure.

Most of the SA+SD dust plumes, as reflected by the hematite mass, were observed at 2-4km, corresponding with the strongest influence of the African easterly jet at this altitude. The SD dust plume was observed throughout the atmospheric column. The dust loading was low within the MBL when transported from the Western African coast. The BC and hematite mass on the ground fell within the range of aircraft measurements in the MBL.

## 4.2 Size distributions

The aerosol total mass and number concentrations between 0.1-0.5µm and 0.5-20µm are shown in Fig. 12. Particles >0.5µm are considered to be mainly composed of dust. An effective diameter is calculated for both the sub-micron (0.1-1µm, $D_{eff,(0.1-1)}$) and super-micron (1-20µm, $D_{eff,(1-20)}$) size ranges for convenient comparison with previous studies. There was a significant contribution of smaller particles within the MBL as reflected by a much higher small particle number, and the smallest $D_{eff,(0.1-1)}$ existed (<0.3µm) when





significantly influenced by SE air masses. These small sub-micron particles are mainly composed of sulphate as shown in Fig. 11. In the dust layer (2-4km), there was a significant increase of large particle number and mass. There was also an increase in $D_{eff,(0.1-1)}$ in the dust layer, which may indicate some of the dust particles contributed to the sub-micron size range.

The $D_{eff,(1-20)}$ was consistently around 5-6μm in the dust layer regardless of air mass origin. This is consistent with the dust layer measured over the western Mediterranean, where the dust had experienced a few days' travel from the Sahara and also showed little vertical variation in $D_{eff,(1-20)}$ (Denjean et al., 2016). This relatively homogenous dust size throughout the vertical profile in the SAL suggests strong vertical mixing or extensive processing over the time scale of transport or a combination of both. In comparison, in the MBL

the particle size showed a wide range of $D_{eff,(1-20)}$, which may reflect the inhomogeneous composition of the layer comprising a mixture of sea-salt and dust particles, and the possibly hygroscopic growth of dust (and other aerosols) in the humid MBL. The dust mass measured at the ground at Praia was at the lower end of measurements in the MBL, due to losses via dry deposition during transport in the MBL.

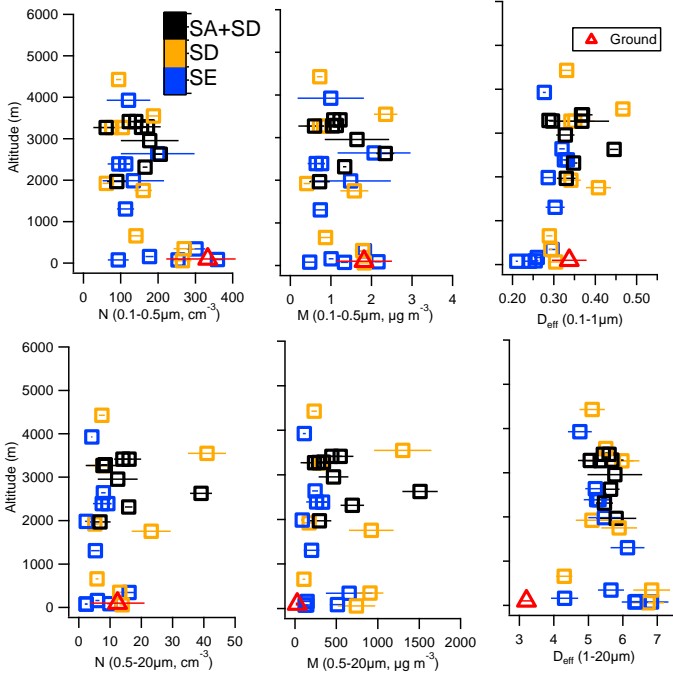

Fig. 12. The particle mass and number concentrations as a function of altitude for particles in the size range d (0.1-0.5)μm and d (0.5-20)μm, and effective diameters in the size range (0.1-1) and (1-20)μm obtained from the PCASP and CDP measurements for all SLRs. The markers are colour coded according to the BT classification and show the mean value together with bars indicating a 1σ variability for each SLR. The surface measurements from the SMPS and APS instruments are also shown (red triangles).





### 4.3 Optical properties

Both the scattering and absorption coefficients were significantly enhanced in the SAL, as shown in Fig. 13. The accumulation mode single scattering albedo (SSA) in the SAL showed some variation and was lower than that in the MBL. The higher SSA at 550nm in the MBL may result from an enhanced contribution from

sea salt particles since there is a reduced dust fraction at this level. The negative value of the scattering Ångström Exponent (SAE) in the SAL suggests the larger particles significantly contributed to the total scattering, and the lack of variation in the SAE, which ranged from -0.4 to -0.2 is in line with the lack of variation in $D_{eff,(1-20)}$ in the SAL and demonstrates the consistent behaviour of the size distribution. In the MBL, the SAE was higher, in particular when the site was influenced by SE air masses. This is consistent

with a higher sub-micron particle concentration in the MBL (Fig. 12) and that in this layer the contribution of total scattering from finer particles such as sulphate is more important; though the absolute value of the SAE is still much lower than for a typical anthropogenic source (i.e. where it is typically >1), which means the scattering was still dominated by the larger particles (Schuster et al., 2006).

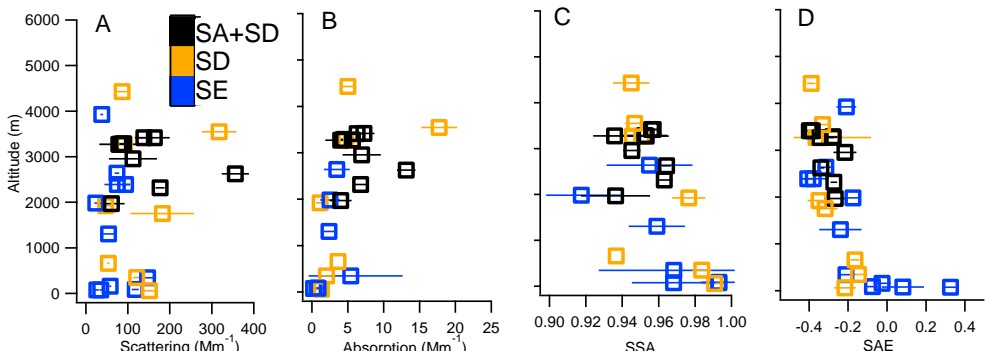

Fig. 13. The optical properties (d<2.5um) for all SLRs. A) Total Scattering (Mm⁻¹); B) Absorption (Mm⁻¹); C) Single Scattering Albedo (SSA at λ=500 nm); D) Scattering Ångström Exponent (SAE) (λ=450 -700nm).

Both BC and the measured hematite contained in the smaller and larger particle size modes, contributes to the light absorption. No obvious correlation between BC mass and SSA was observed (not shown here). The correlation between SSA and the hematite fraction is shown in Fig. 14. There is remarkable anti-correlation

between the measured SSA and hematite volume fraction for the SA+SD air mass-infuenced SAL, suggesting an important contribution of hematite to the absorption, when dust reaching Cabo Verde is transported over the Sahel. The hematite volume fraction was mostly <3.5% for the SD-influenced SAL and the SSA values were always high (~0.94) in this air mass. The MBL generally has a low hematite fraction and high SSA, apart from one SLR with low SSA where the absorption was mainly contributed by BC

(section 6.3).





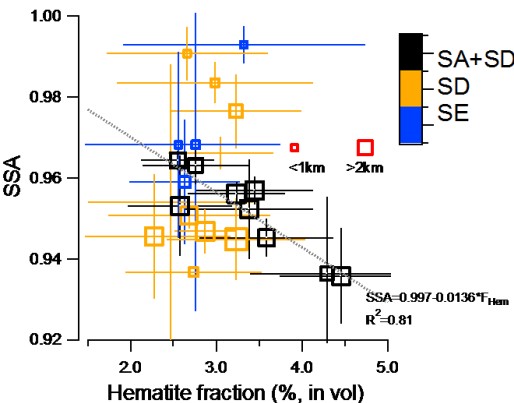

Fig. 14. The correlation between SSA and hematite fraction, the smaller and larger markers denote the measurement in the MBL and SAL respectively.

## 5 The dust aerosol properties as a function of transport time

The dust mass was observed to be higher with shorter transport times, but dropped rapidly and stabilized after 5 days' transport (Fig. 15A). Consistent with the vertical profile, the $D_{eff,(1-20)}$ in the SAL was consistently around 5-6µm and was independent of the dust transport time of 2-6 days. The $D_{eff,(1-20)}$ in MBL (the small markers) shows a decreasing trend with increasing transport time, though the dust transport time in the MBL needs to be treated with caution. The dust size in the measurement region is significantly lower than that measured in the close vicinity of surface sources over the African continent (Ryder et al., 2013b). Previous studies also found that after 1.5 days transport, the particle size, as measured in terms of $D_{eff,(1-20)}$, for Saharan dust became stable (Denjean et al., 2016;Ryder et al., 2013a). For all air mass types, the sub-micron particle $D_{eff,(0.1-1)}$ in the SAL decreased with transport time. In the MBL, the $D_{eff,(0.1-1)}$ was consistently 0.3µm regardless of transport time. For transport times longer than 4 days, $D_{eff,(0.1-1)}$ values converged to 0.3-0.35µm in both the SAL and MBL. By comparing the size distributions as a function of transport time (supplement Fig. S8), we can establish that after 3-5 days of transport about 40±10% of particles above 0.3µm were removed while the small particles showed much lower removal efficiency; after 5-6 days the size distribution stabilized. This may explain the lack of variation of $D_{eff,(1-20)}$ with dust age, since the removal rate over the first few days of transport (leading to significantly reduced total aerosol mass) is high but size-independent, but thereafter there is a reduced deposition rate (with stabilized total aerosol mass).

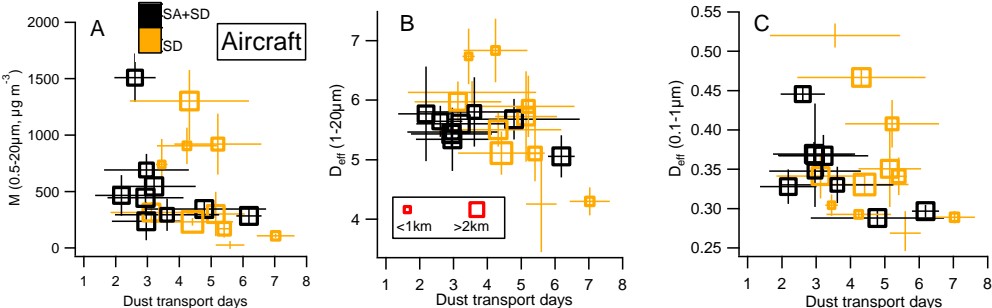

Fig. 15. The aerosol mass and effective diameter as a function of dust transport time for all aircraft SLRs.

The dust mass loading measured at the ground was much lower than from the aircraft, with some evidence of further reduction of aerosol mass (d>0.5µm) with transport time longer than 5 days (Fig. 16A). This may be because the dust was actually being constantly deposited into the MBL from the overlying SAL. The ground site may be constantly influenced by marine air mass and the marine aerosols were not considered to significantly influence the variation of observed size distribution. In contrast to the dust of similar size range in the SAL, the $D_{eff,(1-10)}$ at the ground decreased with longer transport time, at a rate of about 0.1µm/day (Fig. 16B). This may be the result from significant settling of dust to the ground after being processed through the MBL. However, $D_{eff,(0.1-1)}$ showed little variation with transport time at the surface being consistently between 0.3-0.4µm (Fig. 16C), which is at the lower end of the observed $D_{eff,(0.1-1)}$ aircraft values. The almost unvarying particle mass but decreased size for larger particles along transport may result from a combination of gravitational removal of larger particles and re-entrainment of smaller particles from the layers above.

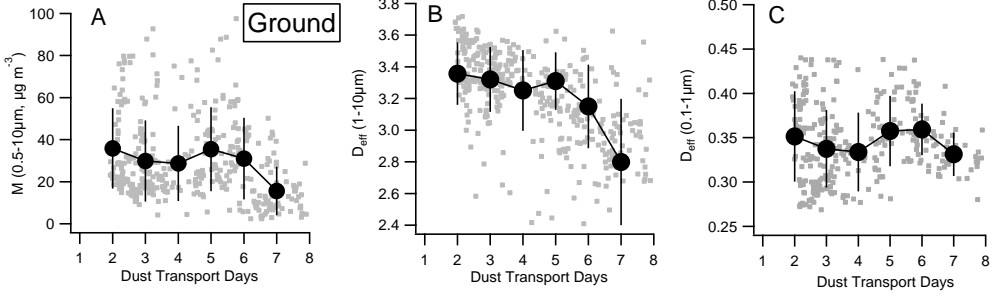

Fig. 16. The particle mass and size as a function of dust transport time for the ground experiment, N.B. $D_{eff}$ (1-10µm) is curtailed at 10µm compared to 20µm in Fig. 15.

Using the measured size distributions and single scattering albedos (SSA), we can derive an SSA-equivalent refractive index by applying a Mie calculation and assuming a homogenous composition throughout the size distribution. This essentially gives the optimum value for the ensemble refractive index based on the Mie-




modelled SSA matching the measurement. The calculations show SSA to be substantially insensitive to the real part of refractive index, and therefore to be consistent with previous studies; a real part of 1.53 is used based on McConnell et al., 2010. The calculation is iterated by step changing the imaginary part of the refractive index ($k$) in steps of resolution of $1e^{-4}$, until the optimum $k$ best matching the measured SSA is
obtained. This calculation is only performed for aircraft SLRs.

Fig. 17A and B show a decreased SSA is measured with longer transport time for the SAL dust, corresponding to an increasing SSA-equivalent imaginary part of the refractive index at 550nm ($k^{550}$). Note that the SSA or $k^{550}$ reported here is only for particle sizes <2.5μm. The MBL aerosol (smaller markers) showed higher SSA and lower $k^{550}$ compared to the SAL. Fig. 17C shows increased hematite fraction with
longer transport for the (SA+SD)-influenced SAL dust, and the (SA+SD) air masses were found to contain a higher hematite fraction compared to SD air masses at the same scale of transport time. This is in general consistent with the hematite distribution in the soil inventory, i.e. the Sahel region has a higher hematite fraction than Sahara, whereas a greater fraction of quartz or feldspar dominates across the central and western Sahara (Nickovic et al., 2012).

Longer transport pathways may bring sources from the far eastern Sahara to the sampling locations. This is consistent with the inventory which identifies soils with a rich content of hematite in these locations. This may partly explain the increased hematite fraction with longer transport. However, considering the significant deposition of dust during such long range transport, the far eastern sources may not have influenced the measurement site significantly. The multimodal size distributions that have been previously
observed close to the sources by off-line SEM analysis of filter samples (Kandler et al., 2007;Kandler et al., 2009): showed quartz, feldspars and calcite to commonly dominate the larger particle sizes while clay minerals tended to have a higher fraction at smaller particle sizes. Absorbing free iron was found to be mainly included in the clay matrix as fine grains from TEM analysis (Jeong et al., 2016). Fig. 17D shows decreased hematite fraction with increased particle size, as represented by $D_{eff,(0.1-2.5)}$, across the detectable
size range of the SP2 for SA+SD air masses. Significantly higher fractions of clay minerals have been observed in the mineral dust aerosol after long range transport, which reflects the changes in particle size distributions due to atmospheric processes such as gravitational settling of larger particles (Kandler et al., 2011a). It should be also noted that goethite commonly coexists with hematite and the absorption may well have contributions from both, though the goethite was not directly measured by the SP2.

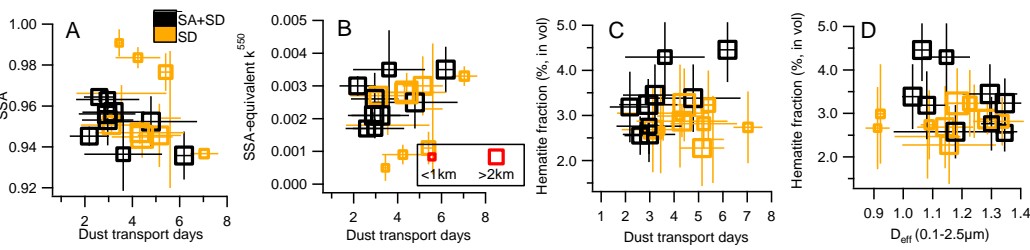





Fig. 17. The aircraft-measured aerosol optical properties A) SSA; B) the imaginary part of refractive index at 550nm $k^{550}$; C) hematite volume fraction as a function of potential dust age; D) hematite volume fraction vs $D_{eff(0.1-2.5)}$. The smaller and larger markers indicate the results in the MBL and SAL respectively.

## 6 Modelling optical properties using size-resolved composition

To evaluate the contribution of absorbing components on the optical properties, the measured BC and hematite fractions combined with the PCASP-measured size distribution (0.1-2.5μm) were incorporated into a Mie calculation. Based on the discussions above, constant BC and hematite fractions (based on the measurements) were applied to the particle size ranges 0.1-0.5μm and 0.5-2.5μm respectively. As the goethite is not directly measured, a constant goethite/hematite mass ratio was assumed. Previous studies reported a goethite/hematite mass ratio of 1-3 (Zhang et al., 2015), the ratio of 1 and 2 was thus chosen to perform a sensitivity test. The calculation was performed at λ=550nm. The refractive index (*n*) of BC was set at 1.85+0.71i (Bond and Bergstrom, 2006), and *n* for hematite and goethite were 3+0.8i and 2.2+0.1i respectively (Bedidi and Cervelle, 1993). Hematite and goethite are commonly dispersed in illite-smectite clay minerals (ISCM) with illite slightly absorbing with a refractive index of $1.53+1e^{-3}i$ (Liao and Seinfeld, 1998). Considering other typical clay compositions such as kaolinite or the other non-clay materials which may be weakly absorbing, a refractive index for the remaining mass of particles with diameters >0.5μm was set to $1.53+1e^{-4}i$. The refractive index apart for the non-BC components in the size range 0.1-0.5μm was set to be 1.50+0i assuming mixed compositions of sulphate and organic (Liu et al., 2015). The absorbing component is deemed to be externally mixed with the others, i.e. without optical interaction causing enhanced absorption due to internal mixing.

Fig. 18 gives a typical example of the modelled absorbing and scattering coefficient contributed from different compositions according to this representation. The absorption of BC dominates at diameters 0.2-0.3μm which is in general consistent with the measured mass distribution of BC. The hematite-goethite absorbs more towards larger size. The main uncertainty introduced here is the assumed constant goethite/hematite ratio which may be source-specific or may vary throughout the size distribution. Based on the measured hematite fraction, applying a ratio of goethite/hematite 1-2 will give a total free iron volume fraction about 4-10%, which is higher than the previous studies from filter samples (Zhang et al., 2015). This may result from a higher free iron fraction in smaller particles given the particle size measured here is up to 2.5μm, which is smaller than particles typically observed on filter samples. Fig. 19 gives a summary of absorption contributions for all SLRs. The absorbing iron contributes 40-80% of the absorption for particles d<2.5μm. The contribution of BC absorption, which ranges from 10-37% in the SAL for particles d<2.5μm, is only significant when the sample contains a considerable fraction of particles d<0.5μm.




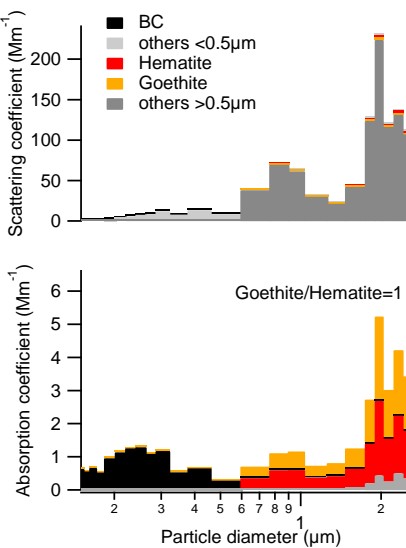

Fig. 18. An example of the scattering coefficient and absorption coefficient as a function of particle size (0.1-2.5μm) for the second SLR during flight B934.

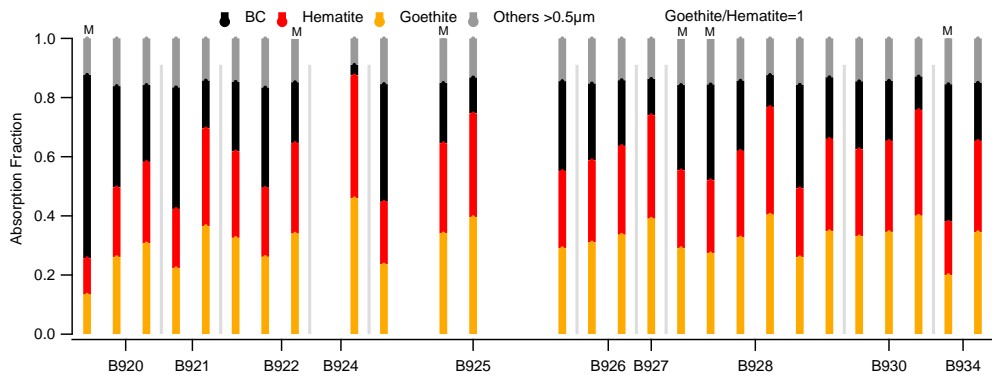

5   Fig. 19. The absorption coefficient fraction (at d=0.1-2.5μm) contributed by different compositions for all SLRs, with the label M indicating the SLRs in the MBL.

The modelled and measured SSA for all SLRs at particle sizes <2.5μm is shown in Fig. 20. The correlation between modelled and measured SSA in the SAL suggests the simple size-resolved, constant composition model used here is a valid approach to reproduce aerosol optical properties. A goethite/hematite mass ratio of

10   2 gives the best estimate of SSA within ±0.02 for all SLRs. There is a significant discrepancy for MBL (small markers) aerosols with the majority of modelled SSA lower than measurement. This may be because the aerosol in the MBL experienced significant deposition since being uplifted from the source, causing the





assumption of constant goethite/hematite ratio to no longer be reliable, or the size distribution of the absorbing components was altered during transport.

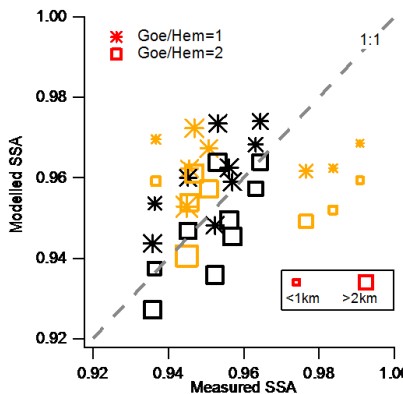

Fig. 20. Modelled and measured SSA for particle d<2.5µm.

## 7 Conclusions

Dust was characterised in the region of Cabo Verde archipelago in Aug. 2015 as the Saharan Air Layer (SAL) transported continental air over the tropical Eastern Atlantic Ocean during summer. The dust loading over Cabo Verde was observed to depend on the strength of the easterly wave. The dust in the SAL or below

in marine boundary layer (MBL) was shown to be transported through different pathways using BT analysis, with the SAL dust transport pathways being either mainly over the Sahara or via significant contact with the Sahel region. Shorter transport and higher mass loading of the SAL dust was generally associated with stronger easterly winds.

The removal rate of aerosols during transport was size dependent. Two effective diameters ($D_{eff}$) were

calculated for the small (0.1-1µm) and large (1-20µm) particles. For the dust in the SAL, $D_{eff,(1-20)}$ was around 5-6µm regardless of transport time, but $D_{eff,0.1-1}$ significantly decreased with longer transport. This is because of a relatively size-independent removal rate at 1-20µm but lower removal efficiency in the sub-micron mode. Although the latter is as expected, gravitational settling would still be expected to increase with size across the larger part of the size spectrum (Weinzierl et al., 2016). As a result the size distribution of the dust

stabilized after a few days' transport time, an observation consistent with a few previous studies (Kalashnikova and Kahn, 2008;Denjean et al., 2016), although application of BTs to determine dust age involves considerable uncertainties since the meteorological reanalyses do not resolve MCS events which drive dust uplift in Saharan summertime. The size distribution in the MBL had higher variability because of the diversity in composition such as with the presence of sea salt and sulphate, and the particles had

experienced significant removal through the MBL.



The absorbing component of dust and in particular the contribution of hematite was measured for the first time by an in-situ method in the 0-2.5 um size range, using aircraft and ground-based measurements. In the SAL, the measured accumulation mode single scattering albedo (SSA) is anti-correlated with the hematite volume fraction when dust has a SA+SD transport pathway, suggesting an important role of the hematite in determining the absorbing properties. For the Sahel-influenced air mass, decreased SSA and increased hematite fraction were observed as the transport time became longer. The longer transport time of dust may have preferentially removed larger particles but retained the smaller clay-iron mixtures. This has consequences for deposition rates of these different particles as a function of air mass history and their effect on biogeochemical cycles, ocean nutrient deposition and aerosol-cloud interactions. The same relationship was not observed for dust solely influenced by SD regions. The imaginary part of refractive index at 550nm ($k^{550}$) computed here ranges from 0.0015 to 0.0035, lower than the value (0.006) archived at the OPAC aerosol database (Hess et al., 1998) which is widely used in the remote sensing communities. This suggests the OPAC may overestimate the absorption of dust, which is also in line with many other studies that that suggest reducing the dust absorption would improve the closure between satellite retrieval and modelling (Balkanski et al., 2007;Chin et al., 2009).

The measured BC and hematite volume fraction combined with the measured size distributions are used to constrain optical properties for particles <2.5μm, by assuming a constant BC and hematite fraction for particles d=0.1-0.5μm and d=0.5-2.5μm respectively. The contributions of each component to the absorbing properties of the aerosol population are evaluated by this model. Given that goethite is not directly measured by our online method, a range of assumed goethite/hematite mass ratios based on literature values were used. We find that a goethite/hematite ratio of about 2 gives the best closure of SSA for the SAL dust. The free iron contributed 40-80% of the absorption in the SAL, but the BC contributed 10-37% depending on its mass fraction in the sub-micron size mode. The dust in the MBL is not well captured by this model most likely due to misinterpretation of the particle composition with high marine influence. Because of the relatively age-independent size of the SAL dust, the optical properties are largely determined by its composition, therefore the change of size-resolved composition with dust age is important in the determination of the radiative forcing effect of dust in this region. As our investigation of optical properties is limited to particles <2.5μm, we cannot rule out changes in the composition of absorbing components beyond this size, as larger particles (up to ~100 um) were observed in this campaign. This is examined in the context of these flights by Ryder et al., (2017). Hence, measurements of larger coarse mode particles are desired for a closure of the entire dust ensemble in future studies.

Data availability



Processed data are available through the ICE-D and AER-D project archive at the British Atmospheric Data Centre (http://badc.nerc.ac.uk/browse/badc). Raw data are archived at the University of Manchester and are available upon request.

Acknowledgements

This work was supported by the Natural Environment Research Council grant Number NE/M001954/1 and the Met Office. Airborne data were obtained using the BAe-146-301 Atmospheric Research Aircraft (ARA) flown by Directflight Ltd and managed by the Facility for Airborne Atmospheric Measurements (FAAM), which is a joint entity of the Natural Environment Research Council and the Met Office. The FAAM flights
in ICE-D and AER-D were funded by the Met Office. The SAVEX-D project was possible thanks to EUFAR TNA (European Union Seventh Framework Programme grant agreement 312609) and projects PROMETEUII/2014/058 and GV/2014/046 from the Valencia Autonomous Government. C. Ryder was funded by NERC grant NE/M018288/1.

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
