# Peer review of "Aircraft and ground measurements of dust aerosols over the West Africa coast in summer 2015 during ICE-D and AER-D"

_Atmospheric Chemistry and Physics, 2017_

## Referee Comment (RC1) · Anonymous Referee #1 · 29 Nov 2017

I read with interest this paper that provides new data from a recent field campaign in Cabo Verde. The novelty of the dataset resides in particular in the inclusion of the SP2 measurements of BC and hematite content in aerosols, which could apport key information to better understand the absorption properties of dust. Also, the size distribution of dust and its changes in link to different aging times and as a function of the source region are analysed. The relation between size, composition, and optical properties is also studied.

The paper is well structured, well written and globally clear. Probably too long and with too many figures in my opinion, but this could be understandable concerning the fact

that many data from an intensive field campaign are described and discussed.

The main conclusions of the paper concern the size distribution and the optical properties of dust : 1. For the size, the paper confirms the changes in the dust size distribution with transport, in agreement with previous analyses close to sources and mid-transport ; 2. For optical properties, it confirms the lower imaginary refractive index for dust compared to the OPAC database, as already evidenced by previous studies, and highligths the importance of having size resolved compositional data to properly retrieve optical properties by Mie calculations.

The paper and the associated dataset deserve publication to ACP. I have only few (mostly) minor comments below :

1. Section 2.3 : I am not a specialist in SP2 measurements and its data analysis, but most of this section is quite unclear to me. I ask the authors to better explain the Fig. 3, as well as the principles, the data analysis and the retrieval procedure from SP2 measurements.

2. Globally, I found that the uncertainites are not well discussed. I encourage the authors to better expalin how uncertainties on measured and retreieved quantities are derived. For instance, on the refractive index or SSA. You consider both SLR variability and measurement uncertainty in your data? How the uncertainties propagate and affect your results and conclusions ?

3. there is a typo in page 8, line 8, probably you mean "smaller range" ?

4. page 12, you use GADS meteorological data for your backtrajectory study. Why not using NCEP reanalyses ? Has this choice an influence on your results ?

5. page 13, line 21-23 : I do not understand what the mass fraction threshold of 5% represents. Could you please better explain the trajectories classification procedure ?

6. Section 5 : I do not find that the discussion on the changes of Deff (0.1-1$\mu$m) with transport days is supported by data. From Fig. 15C I have the impression that Deff is

mostly at 0.30-0.35 $\mu$m independently on the transport days, except from few outliers. I would probably smooth this part of the discussion.

---

## Referee Comment (RC2) · C. H. Twohy (Referee) · 6 Dec 2017

C. H. Twohy (Referee)

twohy@nwra.com

General Comments:

This paper is logical, well written, and provides interesting new results on the properties of Saharan and Sahel dust off the coast of Africa. A variety of types of data are integrated in a meaningful way. It was a pleasure to read and should be published after minor revisions suggested below.

Specific Comments:

Abstract: It would be helpful to convey here that the transport is over both land and

water (and mostly land?), since other studies focus more on transport across the Atlantic. Also, it should be mentioned that the "processing" discussed throughout does not include cloud processing, assuming that is the case (see below).

Line 27-28: Specify if this is true for all cases, or just Sahel.

Page 5, line 24-25: This is an Ice in Clouds Experiment–but clouds are only mentioned as having been screened out. So are the data presented here only representative of dust evolution in clear air? Or were there sometimes clouds upstream that may contribute to dust evolution?

Page 7, lines 22-24: Need more quantitative information/references on this potential enhancement in the Rosemount inlet, as well as the diffusional losses and how they may affect your size distributions.

Figure 2: Horizontal scale on RHS needs minor tick labels.

Page 8, lines 18-20: This was the only confusing part in the paper, and Fig 3 doesn't help much. The mode at larger incandescent signals appears to be at higher Tc, not lower, but maybe I'm misinterpreting what you mean by modes. Perhaps circle modes and/or use arrows? Also, it's not clear what the boxes labelled BC and Hematite are referring to. I think it may be values above and below the horizontal lines, but these lines are almost indiscernible, especially on figure on the right.

Page 9, line 20-21: What do you mean "scaled up"? Corrected for low detection efficiency? Perhaps the discussion on p. 10 about the low detection efficiency at 0.5 microns should be moved up to this section, when you discuss the figure that shows no hematite at small sizes.

Page 11 line 1: Vague; define what you mean by "considerable processing".

Page 13, line 15: Unclear why the number of days over each region is so uncertain. Is it due to trajectory uncertainty, or uncertainty in where the dust originates?

Fig 7D: Mention in legend that vertical bars are standard deviations, assuming this is the case. Also, why are there blank periods in C and D? In cloud?

Fig. 9: Is there a way to specify the number of days over land vs over water, and whether this may have any effect on your results?

Page 19, line 10: Define "low", as low relative to the SAL may not be low in other regions. Page 20, line 9: Define what you mean by "processing".

Page 21: Chen et al, ACP 2011 provided a comprehensive review of SAL optical properties and would be a useful reference here.

Page 21, line 19-20: Nice result.

Fig. 19: For the non MBL cases, it would be nice to also see the primary airmass origin.

Typos:

Page 18, line 8: Composition is misspelled.

Page 21, line 20: influenced is misspelled.

---

## Referee Comment (RC3) · Anonymous Referee #3 · 8 Dec 2017

General: The paper is well written, presents original, new material about dust optical and microphysical properties and is appropriate for ACP.

I have only minor remarks.

Details:

P3, L7-20: Please include references to old Prospero papers, i.e. Carlson and Prospero and Prospero and Carlson from the early 1970ies, when mentioning long-range transport across the Ocean.

P6, L11: Can we have some uncertainty values for all these measurements?

P6, L32-35: Was the field site upwind the run way (north of the runway)? Please provide some information concerning possible contamination of the surface observations by aircraft activity.

P8, Fig. 2: I always like to have date and time of the observations, and also height range of observation. . . in the plot or in the caption.

P9, Fig. 3: Again here, date, time, measurement height in the case of aircraft obs.

P10, L12: . . . particles with diameters > 0.5 $\mu$m .. .. ..

P10, L25: please change to modern units, from mbar to hPa. . .

P11, Fig.4, please again: hPa instead of mba and mbar. . .

P15, L36: uplift. . .? , may be better: emission mechanism

P16 and following pages: Please keep in mind in the discussion that the PBL is an 'open' layer with particle sources in the free troposphere and further contiuoulsy contributing sources from the ground. So, it makes not really sense to me to illuminate the link between dust observations and back-trajectory-based age estimates in days.

P19, Fig. 11: mean values of all flights?

P19, L14-16. Fig 12: Why do you not use the classical fine and coarse mode separation? Fine mode particles with diameters < 1 $\mu$m, coarse mode, all particles > 1 $\mu$m. You separate at 0.5$\mu$m diameter.

P20, Fig. 12: The error bars then show the atmospheric variability (?) or just the uncertainty in the measurements? Please state, preferably in the caption.

P21, Fig 13: Again what do the error bars show?

And when comparing with other observations then please check also profile vs profile observations (e.g. extinction coefficient profiles measured with lidar during SAMUM 2, summer campaign, Praia, Cabo Verde, check Tellus Special Issue on SAMUM2).

[Figure]

**[ACPD](ACPD)**

Interactive
comment

P22, Fig.14, the correlation for SD is bad. . .., again the error bars: what do they show, and what can we conclude when error bars are so large?

P22, L7: Please check AERONET photometer values of r-eff, if available. Are they in good agreement with the aircraft observations?

P23: Discussion of findings, there are always new sources of particles in the PBL, as long as the air mass was over land. . ..why do you then expect trends in D-eff as a function of age?

P23, Fig 16: When seeing Fig 16, I am missing size distribution plots showing fine and coarse dust distributions. What shape does the size distribution have? One mode or bimodal?

P24, Fig 17: Please do not over-interpret the weak or even not existing correlations. It is also confusing that we have sometimes results for D-eff classes from 0.1-1$\mu$m, then 1-20$\mu$m, and here now 0.1-2.5$\mu$m.

Literature needs to be updated.

---

## Author Comment (AC1) · 19 Jan 2018

**Reply for Aircraft and ground measurements of dust aerosols over the West Africa coast in summer 2015 during ICE-D and AER-D**

We thank three referees' important comments which help us to improve the manuscript. In this document, the original comments of referees are in underline, our reply is in normal font, and the corresponding texts in revised manuscript is in red and refer to the new page numbers.

Referee 1:

I read with interest this paper that provides new data from a recent field campaign in Cabo Verde. The novelty of the dataset resides in particular in the inclusion of the SP2 measurements of BC and hematite content in aerosols, which could apport key information to better understand the absorption properties of dust. Also, the size distribution of dust and its changes in link to different aging times and as a function of the source region are analysed. The relation between size, composition, and optical properties is also studied. The paper is well structured, well written and globally clear. Probably too long and with too many figures in my opinion, but this could be understandable concerning the fact that many data from an intensive field campaign are described and discussed. The main conclusions of the paper concern the size distribution and the optical properties of dust : 1. For the size, the paper confirms the changes in the dust size distribution with transport, in agreement with previous analyses close to sources and mid-transport; 2. For optical properties, it confirms the lower imaginary refractive index for dust compared to the OPAC database, as already evidenced by previous studies, and highlights the importance of having size resolved compositional data to properly retrieve optical properties by Mie calculations.

We thank the referee's positive comments and the comprehensive summary of our manuscript.

The paper and the associated dataset deserve publication to ACP. I have only few (mostly) minor comments below:

1. Section 2.3: I am not a specialist in SP2 measurements and its data analysis, but most of this section is quite unclear to me. I ask the authors to better explain the Fig. 3, as well as the principles, the data analysis and the retrieval procedure from SP2 measurements.

   We have improved the explanation of section 2.3 and Fig. 3 for clarity according to referee's suggestions.

   Page 8, line 15, line 20. Page 9, line 6.

2. Globally, I found that the uncertainites are not well discussed. I encourage the authors to better expalin how uncertainties on measured and retreieved quantities are derived. For instance, on the refractive index or SSA. You consider both SLR variability and measurement uncertainty in your data? How the uncertainties propagate and affect your results and conclusions?

We thank the referee to point this out. There are a range of uncertainties for this calculation, such as the refractive index assumption, the size cut off for the inside-cabin measurements, the assumption of constant volume fractions across the size distribution, the uncertainty of the bulk absorption measurement of instrument itself etc. (most of them are not able to precisely defined through the current work), however the major uncertainty for our calculation is found to be the uncertain hematite/goethite mass ratio given the goethite was not directly measured, therefore we performed a sensitivity calculation by varying hematite/goethite mass ratio and conclude this is the major uncertainty for the current calculation we performed.

In the revised version, we have added "The goethite/hematite mass ratio ranging from 1-2 gives an major uncertainty of modelled SSA from 0.7%-1.7%".

Page 27, line 10-11.

3. there is a typo in page 8, line 8, probably you mean "smaller range" ?

yes, thanks for pointing this out and we have revised.

Page 8, line 11.

4. page 12, you use GADS meteorological data for your backtrajectory study. Why not using NCEP reanalyses? Has this choice an influence on your results?

We use the GDAS reanalysis as this is the standard software input for our backtrajectory analysis. To validate the uncertainties of calculated backtrajectory due to using different meteorological reanalysis data is beyond the scope of our study.

5. page 13, line 21-23 : I do not understand what the mass fraction threshold of 5% represents. Could you please better explain the trajectories classification procedure ?

The 5% is an air mass fraction value above which we consider the air mass belongs to a specified type we defined. This value has been varied from 2-10% and we found a value of 5% will best capture the dust event we observed. We have revised related texts to clarify this point.

Page 14, line 5-6.

6. Section 5 : I do not find that the discussion on the changes of Deff (0.1-1_m) with transport days is supported by data. From Fig. 15C I have the impression that Deff is

mostly at 0.30-0.35 _m independently on the transport days, except from few outliers. I would probably smooth this part of the discussion.

We have revised the texts as "For SA+SD air mass type, the sub-micron particle Deff,(0.1-1) in the SAL decreased with transport time, whereas for SD air mass influenced SAL, the Deff,(0.1-1) mostly maintained at 0.32-0.36 μm." according to referee's suggestion.

Page 23, line 13-15.

Referee 2 - C. H. Twohy

General Comments:

This paper is logical, well written, and provides interesting new results on the properties of Saharan and Sahel dust off the coast of Africa. A variety of types of data are integrated in a meaningful way. It was a pleasure to read and should be published after minor revisions suggested below.

We thank the referee's positive comments.

Specific Comments:

Abstract: It would be helpful to convey here that the transport is over both land and water (and mostly land?), since other studies focus more on transport across the Atlantic. Also, it should be mentioned that the "processing" discussed throughout does not include cloud processing, assuming that is the case (see below).

The transport of air mass sometimes could be mainly either over the land or over the sea, to add this statement in the abstract may introduce confusion. According referee's comments, we have added the location information of Praia (where the aircraft took off and the ground-based measurements are) in the abstract to clarify the general location the air masses we experienced.

We have removed the data when we sampled in cloud, however we did not know if the particles we sampled have been cloud-processed or not, to validate if the aerosols have been processed by cloud along the transport pathway may be beyond the scope of our analysis here.

Page 2, line 11.

Line 27-28: Specify if this is true for all cases, or just Sahel.

We have added a word "overall" to clarify this point.

Page 2, line 27.

Page 5, line 24-25: This is an Ice in Clouds Experiment–but clouds are only mentioned as having been screened out.  So are the data presented here only representative   of dust evolution in clear air? Or were there sometimes clouds upstream that may contribute to dust evolution?

Thanks for the referee pointing this out. The ICE-D project is a combined cloud and aerosol project, however this study only focuses on the aerosol results in this project. The in-cloud aerosol measurement is not trustable which we screened out. The uplift of dust was significantly caused by mesoscale convective storms which is mentioned in the main text. As the reply above, the possible cloud processing along the transport pathway may be beyond the scope of this study.

Page 7, lines 22-24: Need more quantitative information/references on this potential

enhancement in the Rosemount inlet, as well as the diffusional losses and how they may affect your size distributions.

We have inserted the following texts according to referee's suggestion:

"The collection efficiency of the standard BAe146 Rosemount inlet has been evaluated by previous studies and was found to have loss or enhancement across the particle size distributions for the measurement sampled from the inlet (Trembath et al., 2012;Ryder et al., 2015)."

Page 7, line 18-20.

Figure 2: Horizontal scale on RHS needs minor tick labels.

We don't understand this referee comment, because Figure 2 already has minor tick marks (and adding minor tick labels would make it too heavy to read).

Page 8, lines 18-20: This was the only confusing part in the paper, and Fig 3 doesn't help much. The mode at larger incandescent signals appears to be at higher Tc, not lower, but maybe I'm misinterpreting what you mean by modes. Perhaps circle modes and/or use arrows? Also, it's not clear what the boxes labelled BC and Hematite are referring to. I think it may be values above and below the horizontal lines, but these lines are almost indiscernible, especially on figure on the right.

We have added the arrows on the right panel of the plot, and changed the color of the separation line to improve the clarity per referee's suggestion.

Revised Fig.3.

Page 9, line 20-21: What do you mean "scaled up"? Corrected for low detection efficiency? Perhaps the discussion on p. 10 about the low detection efficiency at 0.5 microns should be moved up to this section, when you discuss the figure that shows no hematite at small sizes.

We have added more texts to clarify this point, and also reconstructed this section to merge the two parts per referee's suggestion.

Page 9, Line 4-6. Page 10, Line 6-8.

Page 11 line 1: Vague; define what you mean by "considerable processing".

We have revised this as "considerable marine processing".

Page 11, line 9.

Page 13, line 15: Unclear why the number of days over each region is so uncertain. Is it due to trajectory uncertainty, or uncertainty in where the dust originates?

The standard deviation here is not representing the uncertainty but the variation range of the days the air mass has transported over each region. We have revised this part to improve clarity.

Fig 7D: Mention in legend that vertical bars are standard deviations, assuming this is the case. Also, why are there blank periods in C and D? In cloud?

We have added in the revised figure legend: "…with standard deviation representing the variation of transport days for each air mass type. The blank period is when the BTs are located at >5km altitude thus not included in the analysis."

Fig. 9: Is there a way to specify the number of days over land vs over water, and whether this may have any effect on your results?

We have actually performed this analysis, which is denoted as Sea air mass (Fig. 7D blue line shows), this is the time spent over the Atlantic ocean.

Page 19, line 10: Define "low", as low relative to the SAL may not be low in other regions. Page 20, line 9: Define what you mean by "processing".

We have added "relative to the SAL". We find the word "processing" is not accurate here thus delete this.

Page 21: Chen et al, ACP 2011 provided a comprehensive review of SAL optical prop- erties and would be a useful reference here.

The Chen et al., 2011 has been referenced and discussed in the revised version.

Page 21, line 19-20: Nice result.

Thanks.

Fig. 19: For the non MBL cases, it would be nice to also see the primary airmass origin.

We have inserted the primary airmass origin information in Fig. 19.

Typos:

Page 18, line 8: Composition is misspelled.

Page 21, line 20: influenced is misspelled.

Corrected.

Referee 3:

General: The paper is well written, presents original, new material about dust optical and microphysical properties and is appropriate for ACP.

I have only minor remarks. Details:
We thank referee's positive comments.

P3, L7-20: Please include references to old Prospero papers, i.e. Carlson and Prospero and Prospero and Carlson from the early 1970ies, when mentioning long-range transport across the Ocean.
We have inserted the reference per referee's suggestion.

Page 3, line 11.

P6, L11: Can we have some uncertainty values for all these measurements?

The detailed descriptions of uncertainties for absorption and scattering measurements are given in the related references given in the manuscript but we may not extend the discussion of the uncertainties from instruments themselves here.

P6, L32-35: Was the field site upwind the run way (north of the runway)? Please provide some information concerning possible contamination of the surface observations by aircraft activity.

The following texts are added: "The contamination of aircraft emissions was screened out when significant spikes of black carbon concentration measured by the SP2."

Page 6, line 34-35.

P8, Fig. 2: I always like to have date and time of the observations, and also height range of observation. . . in the plot or in the caption.

P9, Fig. 3: Again here, date, time, measurement height in the case of aircraft

obs. P10, L12: . . . particles with diameters > 0.5 $\mu$m .. . ...

These information has been inserted in the figure caption per referee's suggestion.

Revised Fig.2 and Fig.3.

P10, L25: please change to modern units, from mbar to hPa. . .

P11, Fig.4, please again: hPa instead of mba and mbar. ..

Corrected.

P15, L36: uplift. . .? , may be better: emission mechanism

Corrected.

P16 and following pages: Please keep in mind in the discussion that the PBL is an 'open' layer with particle sources in the free troposphere and further contiuoulsy contributing sources from the ground. So, it makes not really sense to me to illuminate the link between dust observations and back-trajectory-based age estimates in days.

We thank referee's suggestion on the dynamic features of the boundary layer. In this study we only generally estimate the boundary height which is about 5km to broadly identify the possible dust source regions over the African continent. We may conduct some detailed boundary layer analysis in the future study.

P19, Fig. 11: mean values of all flights?

The caption has been corrected to state this.

Revised Fig. 11 caption.

P19, L14-16. Fig 12: Why do you not use the classical fine and coarse mode separation? Fine mode particles with diameters < 1 $\mu$m, coarse mode, all particles > 1 $\mu$m. You separate at 0.5$\mu$m diameter.

We have added these texts to clarify: "because in this study particles >0.5µm are considered to be mainly composed of dust (section 2.3) and the 0.5µm is chosen to broadly separate the dust aerosol with the others."

Page 20, line 14-15.

P20, Fig. 12: The error bars then show the atmospheric variability (?) or just the uncertainty in the measurements? Please state, preferably in the caption.

P21, Fig 13: Again what do the error bars show?

It is atmospheric variability and has been stated in the caption.

Revised Fig 12 and Fig. 13 caption.

And when comparing with other observations then please check also profile vs profile observations (e.g. extinction coefficient profiles measured with lidar during SAMUM 2, summer campaign, Praia, Cabo Verde, check Tellus Special Issue on SAMUM2).

The reference Ansmann et al., 2011 is now added and discussed in the revised manuscript.

Page 19, line 7-10.

P22, Fig.14, the correlation for SD is bad. . .., again the error bars: what do they show, and what can we conclude when error bars are so large?

We added: "These indicate the absorbing component in the dust layer is source-dependent but also influenced by the transport mechanism."

Page 22, line 14-15.

P22, L7: Please check AERONET photometer values of r-eff, if available. Are they in good agreement with the aircraft observations?

To validate the remote sensing result is not part of this study but is being prepared in a separate manuscript for the AER-D campaign.

P23: Discussion of findings, there are always new sources of particles in the PBL, as long as the air mass was over land. . ..why do you then expect trends in D-eff as a function of age?

As reply above, we only use the backtrajectory to broadly attribute the possible air mass influence from Sahara or Sahel region, and calculate the overall time spent on each region, but the detailed boundary layer analysis or to use dispersion model to investigate the source potential may be a future work.

P23, Fig 16: When seeing Fig 16, I am missing size distribution plots showing fine and coarse dust distributions. What shape does the size distribution have? One mode or bimodal?

It is bimodal mode which is shown in Fig. 2B.

P24, Fig 17: Please do not over-interpret the weak or even not existing correlations. It is also confusing that we have sometimes results for D-eff classes from 0.1-1$\mu$m, then 1-20$\mu$m, and here now 0.1-2.5$\mu$m.

Literature needs to be updated.

We have modified the texts to smooth the discussion in this section.

Here to choose 0.1-2.5μm is because this is the size cut off for all of the inside-cabin measurement and best explain the measured optical properties as a function of particle size. We have modified the texts to more explicitly explain this point.

Page 25, line 1, line 6, line 14.